# Photoaligned Liquid Crystal Devices with Switchable Hexagonal Diffraction Patterns

**DOI:** 10.3390/ma15072453

**Published:** 2022-03-26

**Authors:** Inge Nys, Brecht Berteloot, Kristiaan Neyts

**Affiliations:** Liquid Crystals & Photonics Group, Department of Electronics and Information Systems, Ghent University, Technologiepark-Zwijnaarde 126, 9052 Ghent, Belgium; brecht.berteloot@ugent.be

**Keywords:** nematic liquid crystal, 2D periodic structures, hexagonal diffraction patterns, photoalignment, out-of-plane reorientation, flat optical elements

## Abstract

Highly efficient optical diffraction can be realized with the help of micrometer-thin liquid crystal (LC) layers with a periodic modulation of the director orientation. Electrical tunability is easily accessible due to the strong stimuli-responsiveness in the LC phase. By using well-designed photoalignment patterns at the surfaces, we experimentally stabilize two dimensional periodic LC configurations with switchable hexagonal diffraction patterns. The alignment direction follows a one-dimensional periodic rotation at both substrates, but with a 60° or 120° rotation between both grating vectors. The resulting LC configuration is studied with the help of polarizing optical microscopy images and the diffraction properties are measured as a function of the voltage. The intricate bulk director configuration is revealed with the help of finite element Q-tensor simulations. Twist conflicts induced by the surface anchoring are resolved by introducing regions with an out-of-plane tilt in the bulk. This avoids the need for singular disclinations in the structures and gives rise to voltage induced tuning without hysteretic behavior.

## 1. Introduction

Thanks to the remarkable combination of fluidity and long-range ordering in the liquid crystal (LC) phase, LC-related research has continued to attract attention over many decades [1,2,3,4]. The long-range ordering of the anisotropic molecules in the LC phase leads to macroscopic anisotropy, with directional dependence of the optical, electrical, magnetic, mechanical, etc. properties. The fluidity, on the other hand, gives rise to strong stimuli-responsiveness, resulting in easily accessible tunability with the help of electric fields, heating, illumination, etc. During the last few decades, high-resolution photoalignment techniques became available to pattern the anchoring of the LC at the confining substrates [4,5,6]. This offers the possibility to steer the self-organization of complex 3D LC configurations in the bulk and in this way engineer devices with desired functionality. In many applications, the LC is used for its electro-optical functionality. The most well-known example is the LC display (LCD), but nowadays LC is also being used to an increasing extent in beam splitters, lenses, diffraction gratings, lasers, beam shaping elements, spatial light modulators, etc. [4,7,8,9,10,11,12,13,14,15,16,17]. Photoalignment is an important enabling technology to design these kinds of flat optical components, where efficient light manipulation is obtained in micrometer-thin LC layers. Molecules in the photoalignment layer typically tend to orient perpendicularly to the polarization direction of the illuminating light, and a well-designed alignment pattern can be imposed by using structured illumination. To do so, different illumination methods have been developed: interference illumination, direct write illumination, plasmonic patterning, structured illumination with the help of a spatial light modulator, etc. [18,19,20,21,22]. The cell can be assembled before or after doing the illumination and in this way different alignment patterns can be imposed at the top and bottom substrate. This offers additional possibilities to design LC components with complex director configurations and advanced optical functionalities.

We here impose a 1D periodically rotating alignment pattern at the top and bottom substrate, with a deviating rotation direction at both substrates (Figure 1). The cells are filled with nematic liquid crystal (NLC) with a positive dielectric anisotropy. The complex alignment configurations at the interfaces give rise to intricate periodic director arrangements in the bulk and electrically tunable 2D diffraction patterns. We investigate the effect of different rotation angles ψ (60°, 90°, 120°) between the top and bottom alignment pattern on the bulk director configuration and on the diffraction characteristics, by comparing experiments with numerical simulations.

A similar alignment configuration has been investigated before, but only for crossed assembly of both substrates (ψ = 90°) [22,23,24,25,26,27] or anti-parallel assembly (ψ = 180°) [28,29]. Somewhat different from the other alignment configurations, ψ = 180° gives rise to a 1D configuration. Regarding the 2D configurations, Crawford et al. in 2005 and Provenzano et al. in 2007 were the first to study the crossed alignment configuration (ψ = 90°) experimentally (in static conditions) [22,23]. Wang et al. used this alignment configuration in 2017 in combination with sliding substrate assembly [24]. We were the first to investigate the director configuration in detail for the crossed assembly of rotating planar alignment patterns [25,26,27]. We demonstrated that surface-induced twist conflicts can be elegantly resolved by introducing a region with vertical director orientation in the bulk, a possibility that was overlooked before. Although the alignment pattern induces planar anchoring at the substrates, the director becomes vertical in a region in the bulk to avoid the creation of disclination lines and minimize the free energy. In order to do so, symmetry breaking takes place, leading to a LC superstructure with a larger period (unit cell 2Λ × 2Λ) than defined by the boundary conditions (unit cell Λ × Λ). In this article, we demonstrate that this concept can be generalized to other (non-90°) rotation angles between the alignment patterns at both substrates. The self-organization of disclination-free LC configurations with local regions with vertical director orientation is demonstrated experimentally and with the help of numerical simulations. The regions with close-to-vertical mid-plane director orientation are approximately lines along a bisector between the axes of director variation on both substrates. The voltage-induced tunability of the director configuration and the diffraction characteristics are also discussed.

## 2. Materials and Methods

### 2.1. Sample Preparation

To prepare the cells, two ITO-coated glass substrates are covered with a photoalignment layer and illuminated with the help of an SLM based illumination setup. As a photoalignment layer, we use Brilliant Yellow (BY, Sigma-Aldrich, St. Louis, MO, USA, 0.2 wt %) dissolved in dimethylformamide (Sigma-Aldrich). After cleaning, the substrates are treated with an ozone-plasma and the photoalignment layer is spincoated (3000 rpm, 30 s). The substrates are subsequently dried on a hotplate for 5 min at 90 °C and the photoalignment procedure is followed before the substrates are glued together. Spherical spacers with 5.5 µm diameter are mixed in the glue (NOA 68) and only the glue edges are illuminated by UV light for curing, while the rest of the cell is protected by aluminum foil. After gluing the substrates together, the cell is filled with nematic LC E7 at 80 °C (above the transition temperature) and cooled down to room temperature. For the photoalignment procedure, a blue laser (Cobolt Twist, 200 mW, λ = 457 nm) is combined with a spatial light modulator (Holoeye Pluto 2) with a resolution of 1920 by 1080 pixels and a pixel pitch of 8 µm. With the help of the necessary quarter-wave plates, the pattern that is displayed on the SLM defines the (linearly polarized) polarization pattern projected on the substrate. The setup itself is discussed in more detail in previous articles [16,20].

After illumination, the substrates are assembled so that there is a ψ rotation between the 1D rotating alignment pattern at the top and bottom substrate. The azimuthal angle φ w.r.t. the *x*-axis is varying at the substrates as
(1)$$\varphi \left(r\right)={\;\varphi }_{0}-K.r\;K_{t,b}=\frac{\pi }{\Lambda }\cos\frac{\psi }{2}\;1_{x}\;\pm \;\;\frac{\pi }{\Lambda }\sin\frac{\psi }{2}\;1_{y}$$
with **K**_t_ and **K**_b_ respectively being the grating vector at the top and bottom substrate. The alignment period Λ was equal to 10 µm. To obtain alignment configurations with different ψ in the same cell, the alignment patterns imposed by the SLM are rotated over the appropriate angle for different illuminated areas. In our convention, the alignment patterns are rotated over ±ψ/2 around the *z*-axis for the top and bottom substrate (Figure 1). The bisectors between the rotated axes are along the *x*- and the *y*-axis. With these definitions, we find linear regions parallel to the *x*-axis in which the top and bottom alignment are parallel, indicated by yellow and green in Figure 1. The boundary conditions are periodic in the *y*-direction with period Λ_y_
(2)$$\Lambda_{y}=\frac{\Lambda }{\sin\left(\frac{\psi }{2}\right)}$$
and in the *x*-direction with period Λ_x_
(3)$$\Lambda_{x}=\frac{\Lambda }{\cos\left(\frac{\psi }{2}\right)}.$$

### 2.2. FE Q-Tensor Simulations and Simulations of Optical Microscope Images

To simulate the 3D director configuration in the bulk of the cell, finite element Q-tensor simulations are used [30,31]. The representation of the LC in terms of the Q-tensor—combining information about the LC order and orientation—can simulate configuration with and without singular disclinations. The simulation program is looking for a stationary solution that minimized the Landau De Gennes free energy, being the sum of the elastic distortion energy, electric energy (when a voltage is applied), surface energy, and thermotropic (or Landau) energy [32,33]. The energy contributions are expressed in terms of the Q-tensor and its spatial derivatives and material parameter of the LC E7 are taken into account (K_11_ = 11.1 pN, K_22_ = 6.5 pN, K_33_ = 17.1 pN, ε_perp_ = 5.2, ε_para_ = 19). The values of the bulk thermotropic coefficients A, B, and C (A = −174 N/m^2^, B = −2120 N/m^2^, C = 1740 N/m^2^) are based on the experimentally measured values for 5CB at a reduced temperature of −2 °C [34]. Smaller values for these constants are used to increase the natural length scale of variations in the order parameter and to obtain faster numerical convergence [35]. A tetrahedral mesh is used to describe the volume of the LC and a voltage can be applied between the z = 0 and z = *d* coordinate (uniform electrodes at the substrates). Periodic boundary conditions are used at the edges of the simulation domain for constant *x*- or *y*-coordinate, and fixed anchoring is imposed at the substrates z = 0 and z = *d* = 5.5 µm. Since we can only impose periodic boundary conditions for a rectangular unit cell, with edges for a fixed *x*- and *y*-coordinate, we simulated a rectangular unit cell as shown in Figure 1.

Based on the simulation results for the director configuration, optical simulations for the microscopy images are performed with the help of the open source software Nemaktis (https://github.com/warthan07/Nemaktis (accessed on 25 March 2022)). This software implements part of the generalized beam propagation method for birefringent media described by Poy et al. and can include the focusing optics of the microscopy and the spectrum of the illuminant [36]. Simulations are done for 15 different wavelengths equally distributed over the visible wavelength range and the spectrum of a CIE. An illuminant is used to weight the different contributions and obtain a color image.

## 3. Experimental Results

### 3.1. Microscopy Images

Experimentally measured polarizing optical microscopy (POM) images are shown in Figure 2 and Figure 3 for gratings with a different rotation angle between the top and bottom substrate. Results without applied voltage are shown in Figure 2 (for ψ = 60°, 90°, 120°) while measurements for different applied voltages are shown in Figure 3 (for ψ = 60° and 120°). The frequency of the applied electric field is 1 kHz. The simulated unit cell is indicated in the microscopy images. In all cases, a disclination-free director configuration is observed in the bulk of the cell. The structure smoothly varies when a voltage is applied and no hysteresis behavior occurs.

The microscopy images immediately show that similarities exist between the alignment configurations with different rotation angles ψ. Although the dimensions of the unit cell are stretched for different alignment configurations, horizontal bands of darker and brighter appearance can be clearly recognized in all images (Figure 2). The spacing along the *y*-direction becomes smaller for increasing ψ. The dark bands correspond to the regions with (anti-)parallel alignment at the top and bottom substrate as indicated in Figure 1. Within the simulated area with $2\Lambda_{x}\times 2\Lambda_{y}$ dimension, four brighter and four darker horizontal bands appear. Without applied voltage, a clear distinction can be seen between two types of dark bands, occurring alternatingly. One of them is wider and consists of dark and slightly more bright regions next to each other (along *x*) with little variation. The other type is shifted along the *y*-direction over a distance $\Lambda_{y}$/2 and contains a thin dark line. The different bright horizontal bands (shifted along the *y*-direction over a distance $\Lambda_{y}$/2) are less easily distinguished at 0 V. For slightly increased voltages the distinction becomes more clear and the unit cell can be better recognized (Figure 3). The exact optical appearance and the observed colors depend on the alignment configuration and cell thickness.

At sufficiently high voltages (>6 V_pp_), the symmetry in the POM images increases and a grid of bright points is observed (Figure 3). The director is oriented close-to-vertically in the bulk and appreciable deviations from the vertical direction are only present close to the photoaligned interfaces. Between crossed polarizers, no light is transmitted when the azimuthal anchoring at the top and bottom substrate is in perpendicular directions. Bright areas in the POM images correspond to regions where the director at the top and bottom substrate is parallel and under an angle (of ~45°) with respect to the crossed polarizers. For the tested alignment configuration, the lines with parallel top and bottom alignment are running parallel to the *x*-axis, with a spacing along the *y*-direction equal to $\Lambda_{y}$/2 = Λ/(2 sin(ψ/2)). The spacing between bright points along the *x*-axis is $\Lambda_{x}/2$ = Λ/(2 cos(ψ/2)). For ψ = 90°, as studied before [25], the structure becomes centro-symmetric at high voltages with a periodic pattern along the *x*- and *y*-axis with a Λ/√2 periodicity in both directions).

### 3.2. Diffraction Measurements

The intricate 2D alignment configuration, composed of 1D periodically rotating alignment patterns at both substrates, gives rise to interesting diffraction behavior. The working principle can be understood in terms of the geometric phase or so called Pancharatnam–Berry phase of light [37,38]. Very efficient geometric phase gratings, working in transmission or reflection, have been demonstrated before [11,12,13,14]. The focus is often on 1D gratings or lens configurations, but here we investigate 2D gratings with different symmetry. The voltage-dependent diffraction characteristic was measured for an alignment configuration with ψ = 60° and 120° and important features are explained. These particular rotation angles between the alignment patterns at the top and bottom substrate give rise to diffraction patterns with hexagonal symmetry.

Experimentally observed diffraction patterns for different applied voltages are shown in Figure 4 for ψ = 60° (a,b) and ψ = 120° (c). The incident light from a helium-neon laser (λ = 633 nm) was circularly polarized in (a) and linearly polarized along the *y*-axis in (b) and (c). A polarizer was used to improve the degree of linear polarization and an additional quarter-wave plate was inserted for the experiments in Figure 4a. A circular area with a diameter of approximately 1 mm was illuminated and the incident beam power was 1120 µW. The light was captured on a screen behind the LC cell (in transmission) and images of the screen were taken with a smartphone camera. The cell was mounted so that the incident light beam first hits the top substrate and leaves the cell through the bottom substrate. For these geometric phase gratings, the diffraction pattern is asymmetric (with respect to inversion of k_x_ and k_y_) for circularly polarized incident light. For incident light with the opposite circular polarization the same results are obtained but rotated over 180 degrees. For a plane monochromatic wave incident on a purely periodic structure, the projection of the wave vector on the *xy*-plane for each diffracted beam is given by
(4)$$k_{xy}\left(i,j\right)=i\;2K_{t}+j\;2K_{b}$$
with **K**_b_ and **K**_t_ being the vectors indicating the rotation of the photoalignment at the bottom and top substrate as defined in Equation (1). The indices *i* and *j* indicate the order of diffraction. Diffraction order (−1, 0) corresponds with diffraction in the −1st order associated with the top substrate, while (0, −1) corresponds to −1st order diffraction associated with the bottom substrate (Figure 4d). Because the period of the bulk configuration is larger than the period at the substrate, indices with halve integers are also found in the diffraction pattern (Figure 4).

The symmetry of the alignment configuration can clearly be recognized in the diffraction patterns (Figure 4). The distribution of light into the different orders is rather complex and non-monotonous behavior is observed as a function of the voltage since the cell is relatively thick (Δnd/λ = 0.2 × 5.5 µm/0.633 µm >> 1/2). For the experiments shown in Figure 4a, the experimentally measured power into the most important diffraction orders is summarized in Figure 4e. The diffraction order that is strongest at 0 V is diffracted towards the negative k*_y_*-axis. This order (denoted (−1/2, 1/2) in Figure 4d) first decreases for increasing voltages, but shows an additional maximum around 4 V_pp_ before going to zero at high voltages. The zero-order transmission is small at low voltages but becomes dominant at high voltages.

## 4. Simulation Results

To understand how the LC director is oriented in the bulk of the layer, numerical simulations are performed. The simulated area is a rectangle with dimension $2\Lambda_{x}*2\Lambda_{y}\;$ = 8 × Λ^2^/sinψ (Figure 1 and Figure 2). This is two times larger than the unit cell of the bulk LC configuration (Figure 2), since our simulation tool can only impose periodic boundary conditions in a rectangular volume with bounding plates parallel to the coordinate planes. In the rectangular unit cell with basis $2\Lambda_{x}$ and height $2\Lambda_{y}$ periodic boundary conditions apply (Figure 1).

In Figure 5 and Figure 6, the simulation results are summarized for different applied voltages for ψ = 60° and ψ = 120° respectively. The details of the director configuration depend on the parameters Λ, d, and ψ, but some general observations can be made. When observing the mid-plane cross-section (Figure 5a and Figure 6a) it is clear that alternating regions with high and low tilt angle appear along the *y*-axis with period $\Lambda_{y}$. In the simulated area, two line-like regions appear with close-to-vertical mid-plane director orientation at 0 V for y = $\Lambda_{y}$/2 and y = 3$\Lambda_{y}$/2. This corresponds to positions in the alignment configuration where the director at the top and bottom substrate is anti-parallel as shown in Figure 1. In between, for y ≈ 0 or y ≈ $\Lambda_{y}$, the alignment at the top and bottom substrate is parallel (Figure 1) and the director in the bulk remains roughly parallel to the substrates (Figure 5 and Figure 6), with the same azimuthal angle as defined at the substrates. Some deviation from the planar alignment in the bulk is observed in these regions, with modulation of the tilt-angle along the *x*-direction (Figure 5d and Figure 6d). This phenomenon was also observed before in cells with ψ = 90° and the deviations from the planar orientation are in this case related to the Λ/*d* ratio. Decreasing Λ/*d* leads to increasing deviations from the horizontal mid-plane alignment and an increased asymmetry in the unit cell [25,26,27]. The fact that these line-regions at y ≈ 0 or y ≈ $\Lambda_{y}$ appear rather dark in the POM images at 0 V (Figure 2, Figure 5b, and Figure 6b) is linked to the retardation in the LC layer. By also looking at the results for other voltages, the regions with azimuthal orientation along one of the polarizers (dark) or under ~45° angle w.r.t. the polarizers (bright) can be better recognized.

## 5. Discussion

In general, very good agreement is observed between experimental POM images (Figure 2 and Figure 3) and simulated microscope images (Figure 5b and Figure 6b). This agreement proves that the complex bulk director configuration is well-understood. Minor changes between both can be related to small changes in the exact cell thickness, illumination conditions, imperfections in the SLM photoalignment illumination, and focusing effects in the microscope. Moreover, strong anchoring is assumed in the simulations while finite anchoring strength will slightly influence the optical properties.

For the studied alignment configurations, local out-of-plane reorientation of the director can resolve the twist-conflicts induced by the substrates. In order to do so, the unit cell for the bulk director configuration needs to be enlarged with respect to the alignment configuration at the substrates. The unit cell for the alignment configuration has area $\frac{1}{2}\times \Lambda_{x}\times \Lambda_{y}$ (Figure 1) while the unit cell of the bulk structure has a parallelogram shape with area $2\times \Lambda_{x}\times \Lambda_{y}$. Neighboring horizontal lines spaced over $\Lambda_{y}$/2 in the *y*-direction, with equal top and bottom alignment, correspond with another bulk director orientation. This can be intuitively visualized by representing the alignment pattern with the help of a vector, having a head and tail (Figure 1). Although the alignment pattern itself imposes planar anchoring without pretilt, out-of-plane reorientation is present in the bulk. Therefore, regions with parallel or anti-parallel surface orientation of the vector are non-equivalent. In the bulk, these two-line regions (e.g., at y = 0 and y = $\Lambda_{y}$/2) behave differently, with an alteration of line regions with approximately parallel director and line regions with close-to-homeotropic orientation in the mid-plane. These observations are similar to what we observed before in cells with crossed assembly (ψ = 90°) of the alignment patterns [25,26,27]. Although the details of the director configurations are obviously different, the general concept behind the structural formation in the bulk can be generalized. Local out-of-plane director orientation in the bulk can resolve twist conflicts at the aligning substrates for all tested rotation angles ψ.

The fact that symmetry breaking takes place to form the bulk LC configuration is inherently linked to the existence of multiple equivalent solutions within the unit cell. In an experimental device, different shifted domains can exist. A disclination line is present between two shifted domains, as can be seen in Figure 7. These borders between the domains break the periodicity and lead to some background of light between the integer (or half integer) diffraction orders. In general, the fact that the boundary conditions can lead to multiple equivalent bulk structures offers potential for multi-stable devices that can be switched electrically or optically with low power consumption [28,39,40].

It is worth noting that for a similar alignment configuration with ψ = 90°, Wang et al. reported the formation of a web of twist-disclination lines in the bulk when using sliding substrate assembly [24]. To avoid the need for these kinds of singular disclinations in the structure, regions with a close-to-vertical mid-plane director orientation are necessary. In our experiments with fixed substrates (with *d* = 5.5 µm and Λ = 10 µm) we did not observe a web of disclination lines for any of the studied alignment configurations (ψ = 60°, 90°, 120°). A defect-free director configuration, with regions with vertical director orientation, is naturally favored after going through a voltage cycle (shortly applying a high voltage). In our experiments, this disclination-free configuration was stable for months and this structure was also formed immediately after cooling from the isotropic phase without applied voltage. This is different from the results reported by Honma et al. in a cell with large alignment period Λ = 80 µm and rotation angle ψ = 180° (1D configuration). In this case, bistable switching between different configurations (with and without singular disclinations) was observed [28].

The patterned structures with ψ = 60° or 120° give rise to hexagonal diffraction patterns with easy electrical tunability. The application of a small voltage between the uniform electrodes at the top and bottom substrate can redistribute power into different orders. Numerical simulations are necessary to predict the diffraction efficiency and its voltage dependence (see [25] for ψ = 90°), but the general concepts can be understood from a theoretical basis as outlined below.

The behavior of 1D gratings with the same periodically rotating alignment pattern at the top and bottom substrates, is theoretically well-described. For 1D LC polarization gratings working in transmission, the diffraction efficiency can be estimated theoretically based on the paraxial approximation for an infinite grating with $2\pi \lambda d/\left(n_{0}\;\Lambda^{2}\;\right)\ll 1$ [41,42]. For circularly polarized input light, a very high (theoretically 100%) efficiency into the first diffraction order is obtained when the half-wave condition is satisfied $\frac{\Delta nd}{\lambda }=1/2$. Left- and right-handed circularly polarized light is diffracted in opposite directions (positive and negative diffraction order) and linearly polarized light is equally diffracted in both directions. The polarization state is conserved in the zeroth order while the handedness of the incident circular polarization is inversed for diffraction into the first order. This also holds when the half-wave condition is not satisfied.

At sufficiently high voltages, the 2D gratings studied here can be seen as a combination of two rotated 1D gratings that are decoupled by a layer of *z*-oriented LC in the bulk. The region with vertical director orientation in the middle of the cell (at high voltages) can approximately describe the behavior based on the two gratings associated with the top and bottom substrate respectively. Therefore at high voltages, the theoretical description presented above (for 1D gratings) can be used to understand the diffraction in the first diffraction orders (±1, 0) and (0, ±1), associated with the top and bottom substrate, and in the zero order (0, 0). The amount of light that is diffracted into the different orders depends on the voltage via the retardation. For vanishing retardation (at high voltages) light is concentrated in the zero order, with a polarization that is conserved. For circularly polarized incident light, light diffracted in the first diffraction orders is circularly polarized with a handedness that is inverted. These diffraction spots, associated with first order diffraction at the top and bottom grating respectively, are observed in position (−1, 0) and (0, −1) respectively. As can be seen in Figure 4a, a small amount of light is also diffracted towards (0, +1). This side effect might be related to small changes in the light polarization upon propagation through the LC layer, before reaching the grating associated with the bottom substrate. Diffraction into the (−1, +1) order, as can be seen in Figure 4 (at higher voltages), results from a combined effect from the grating at the top and bottom substrate: since the handedness is inversed by the diffraction towards the −1st order associated with the top substrate, the light can be diffracted a second time towards the +1st order associated with the bottom substrate. Thanks to this combined effect, the light diffracted into the (−1, +1) order has again the same polarization as the incident light. This diffraction spot is oriented along the k*_y_*-axis (Figure 4).

The other diffraction spot (−1/2, +1/2) that appears along the k*_y_*-axis, is especially important at lower voltages. The (−1/2, +1/2) diffraction order is dominant at 0 V and results from the $\Lambda_{y}$ periodicity along the *y*-axis that is present at 0 V (Figure 2 and Figure 3). As can be observed in the POM images in Figure 3, a structure with period $\Lambda_{y}$ along the *y*-axis is formed at 0 V while the observed period at high voltages is $\Lambda_{y}$/2. This intuitively explains the presence of a strong diffraction order (−1/2, +1/2) along the k*_y_*-axis at low voltages. To quantitatively describe the distribution of light into the different orders, detailed numerical simulations are necessary.

## 6. Conclusions

The diffraction characteristics and the observed microscopy images can be well understood with the help of numerical simulations for the director configuration. The formation of a 2D LC superstructure without disclinations is driven by energy minimization. Twist conflicts in the confining substrates are resolved with the help of out-of-plane director reorientation in the bulk of the layer and this concept is generally applicable for different rotation angles ψ between the alignment patterns at both substrates. Alternating line regions with (anti-)parallel top and bottom alignment lead to bulk regions with respectively close-to-vertical and roughly planar director orientation. Rotation angles ψ = 60° and 120° give rise to hexagonal diffraction patterns that can be reversibly switched with the help of small electric fields.

## Figures and Tables

**Figure 1 materials-15-02453-f001:**
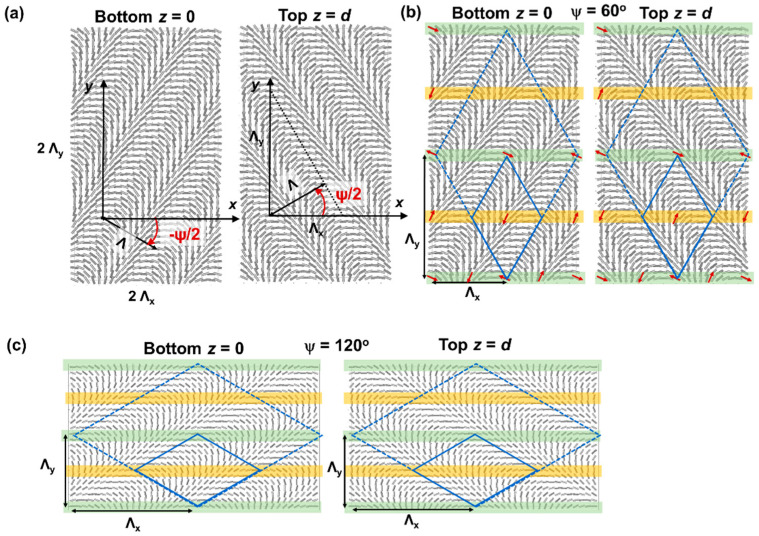
Alignment configuration at the bottom (*z* = 0) and top (*z* = *d =* 5.5 µm) substrate for ψ = 60° (**a**,**b**) ψ = 120° (**c**). One simulated unit cell is shown. Periodic boundary conditions are applied at the edges with a fixed *x*- or *y*-coordinate. (**b**,**c**) Indication of regions with parallel (green) and anti-parallel (yellow) director alignment at the top and bottom substrate when a vector representation of the director is used. The unit cell for the bulk LC director configuration is shown in dashed lines, together with the unit cell for the alignment pattern in full lines.

**Figure 2 materials-15-02453-f002:**
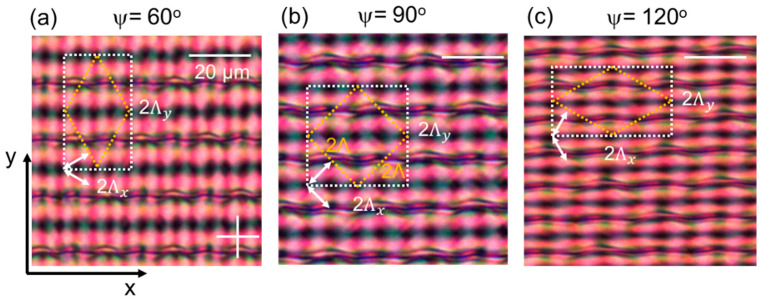
Experimental polarization optical microscopy images for gratings with ψ = 60°, 90°, 120°, from left to right (**a**–**c**). The polarizer and analyzer are crossed and oriented along the *x*- and *y*-axes respectively. The grating vectors at the top and bottom substrate are indicated with white arrows. The simulated unit cell, with dimension $2\Lambda_{x}\times 2\Lambda_{y}$, is shown in a white dotted rectangle while the unit cell for the bulk director configuration is indicated in yellow.

**Figure 3 materials-15-02453-f003:**
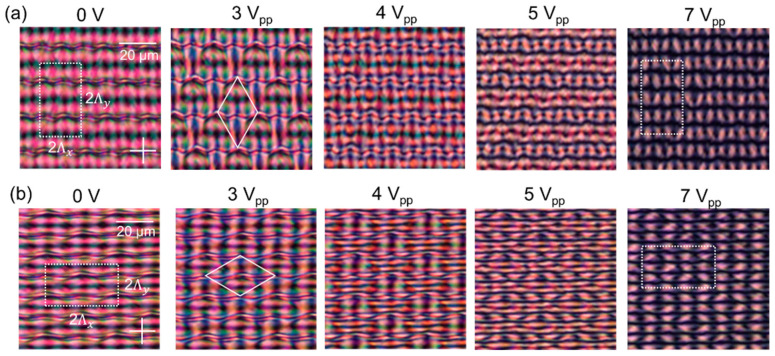
Experimental polarization optical microscopy images for gratings with ψ = 60° (**a**) ψ = 120° (**b**). The applied voltage is increasing from left (0 V) to right (7 V_pp_). The polarizer and analyzer are crossed and oriented along the *x*- and *y*-axes respectively. The simulated unit cell, with dimension $2\Lambda_{x}\times 2\Lambda_{y}$, is shown in a white dotted rectangle (for 0 V and 7 V_pp_) and the bulk unit cell is shown in a white diamond (for 3 V_pp_).

**Figure 4 materials-15-02453-f004:**
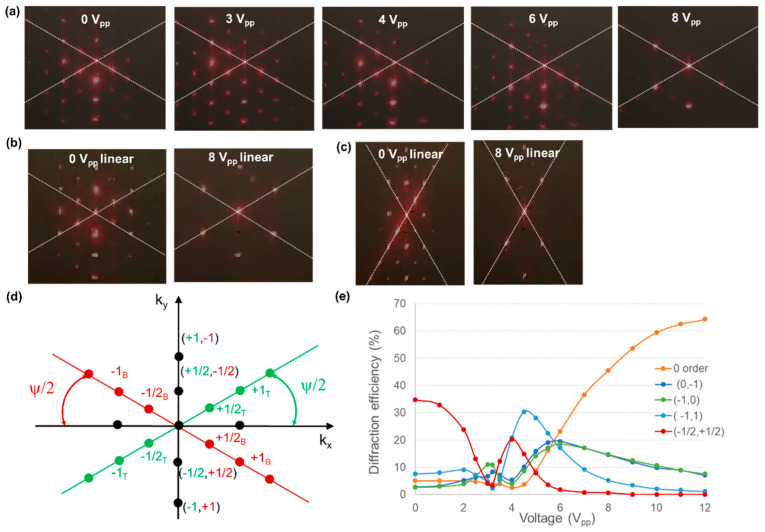
Experimental diffraction images at different voltages for a grating with ψ = 60° (**a**,**b**) and ψ = 120° (**c**). The incident light was circularly polarized in (**a**) and linearly polarized in (**b**,**c**). A schematic representation of the diffraction orders for ψ = 60° is shown in (**d**). The measured power in four different diffraction orders is shown in figure (**e**) for ψ = 60° and circularly polarized incident light (corresponding to experiment (**a**)).

**Figure 5 materials-15-02453-f005:**
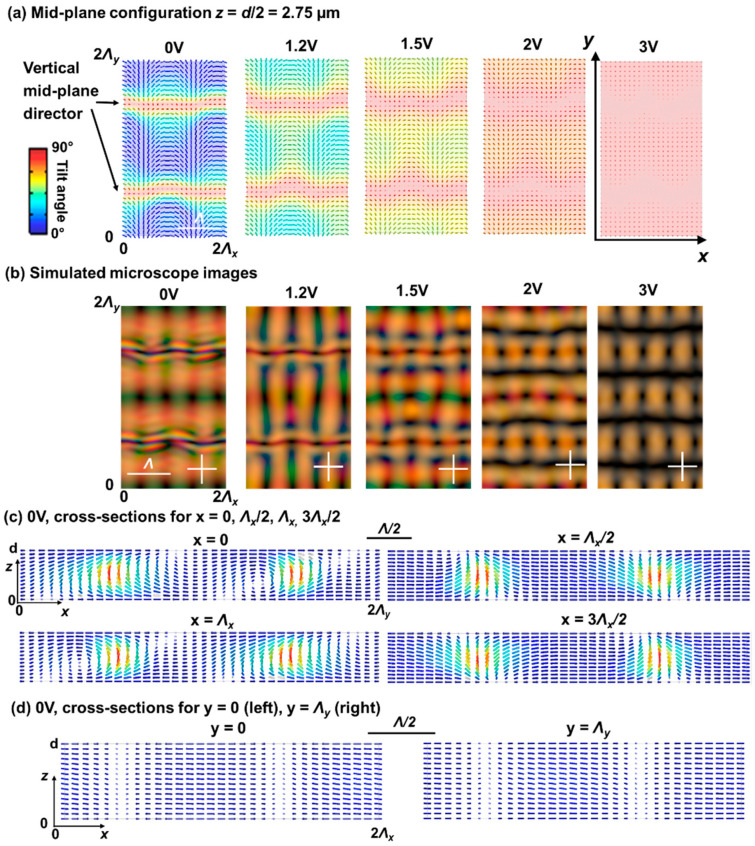
Simulated director configurations (**a**,**c**,**d**) and transmission images (**b**) for ψ = 60°. The mid-plane (*z* = *d*/2 = 2.75 µm) director configuration is shown in (**a**) for different applied voltages (root mean square values). Cross-sections for constant *x*- and *y*-coordinate are shown in (**c**,**d**) at 0 V. The color of the director indicates the tilt angle with respect to the *xy*-plane. The alignment period Λ is 10 µm.

**Figure 6 materials-15-02453-f006:**
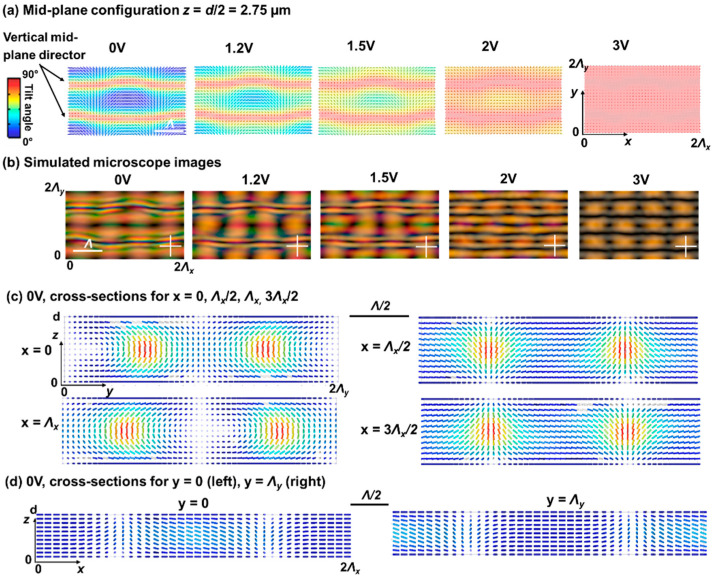
Simulated director configurations (**a**,**c**,**d**) and transmission images (**b**) for ψ = 120°. The mid-plane (*z* = *d*/2 = 2.75 µm) director configuration is shown in (**a**) for different applied voltages (root mean square values). Cross-sections for constant *x*- and *y*-coordinates are shown in (**c**,**d**) at 0 V. The color of the director indicates the tilt angle with respect to the *xy*-plane. The alignment period Λ is 10 µm.

**Figure 7 materials-15-02453-f007:**
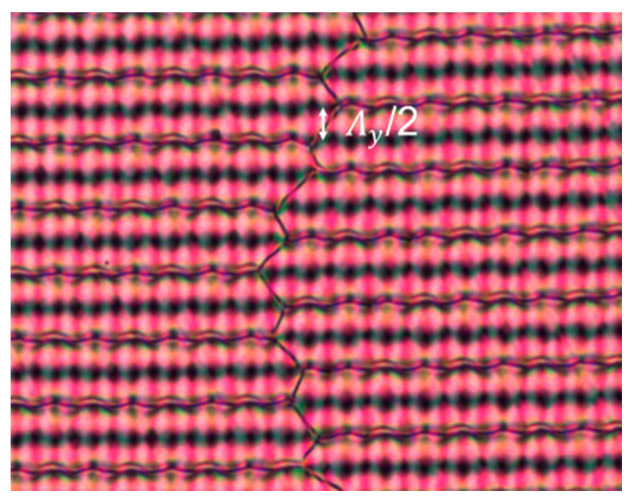
POM image showing the existence of two shifted domains separated by a disclination line. ψ = 60° in this example.

## Data Availability

Data may be provided by the authors upon request.

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
