# Peer review of "Photoaligned Liquid Crystal Devices with Switchable Hexagonal Diffraction Patterns"

_materials, 2022, doi:10.3390/ma15072453_

Round 1
Reviewer 1 Report
The manuscript entitled “Photoaligned liquid crystal devices with switchable hexagonal diffraction patterns” reports the formation of 2D periodic LC configurations without disclinations by using well-designed photoalignment patterns at the surfaces with alignment direction following a 60° or 120° rotation. By applying low voltages, the switchable hexagonal diffraction patterns are obtained. The authors give detailed descriptions in the experimental results and numerical simulations, which agree well with each other to prove complex bulk LC director configurations. The work can be considered for the publication in this journal after revising the following few issues:
- Page 5, Fig. 2: in the caption, “the unit cell for the bulk director configuration is indicated in yellow for ψ = 90° (c).” This sentence should correspond to (b).
- Page 6, line 227: “Equation 1” is not indicated in the manuscript. Please indicate the serial number of all equations.
- Page 6, line 231: please delete the “of” in the sentence.
- Fig. 4(d) and 4(e): they are too small. Please provide large and clear figures with high resolution.
- In Fig. 5(c) and Fig. 6(c), x = 2Λx/3, should be x = 3Λx/2?
- Page 9, line 297: simulated microscope images should correspond to Fig. 5(b) and Fig. 6(b), not Fig. 7, 8.
- Page 10, Fig. 7: should not have (a) and the content in line 325 in the same page should also be revised.
Reviewer 2 Report
The manuscript by Nys et al. reports an interesting optical property of a liquid crystal grating fabricated by combining two substrates having a 1D periodic director pattern enabled by a phot alignment technique. The structure of the director field and the effect of electric field are clearly given. The diffraction properties are also studied in the limited cases. Although the theoretical explanation is not given, the results are qualitatively discussed. These can be a useful information for applied research using photo alignment techniques. The paper can be published after addressing some minor points.
I feel that the explanations for the diffraction measurements are not sufficient and need to be modified. The contrast of images in figures 4(a)-(c) is not good. Are these diffraction patterns captured with a camera or at a screen? In addition, the light intensity at the spots is not symmetric for example along kx and ky. Is this an essential result of this system or not? I also suggest that basic information be included (for example, the power of incident laser beam, the beam size, and optical setups). The vertical axis of Figure 4(e) can be written with the diffraction efficiency. How much area of the sample was used for the diffraction experiments? The direction of polarization of incident beam should be added for Figures 4(b),4(c)
The frequency of the electric field should be added.
Line 226 (page 6): the rotation speed could be “the rotation angle”
Line 227 (page 7): Equation 1 is not indicated.
Line 297 (page 9): Fig. 7, 8 should be Fig. 5, 6.
Reviewer 3 Report
Report on the manuscript entitled: “Photoaligned liquid crystal devices with switchable hexagonal 2 diffraction patterns” by Inge Nys et al.
The authors present a complete study of the characteristics of a type of switchable diffraction pattern with hexagonal symmetry. The device consists of a liquid crystal cell were 1D periodic patterns have been recorded on both ITO-coated glasses by means of a photoalignment technique and rotated 60º and 120º respect to each other. A voltage (up to 8 Volts) across the cell controls the orientation of the nematic molecules in the bulk achieving different disclination-free 3D configurations that are studied by means of polarization optical microscopy and light diffraction, and also compared with numerical simulations. This is, in my opinion, a very interesting work in the line of previous ones published by the same authors (Refs. 25 – 27 in the manuscript). The experimental results are of a very good quality and I agree, in general, with the interpretation given for the different results. Therefore, I consider that the manuscript is interesting and fulfills the requirements for publication in MATERIALS.
There is, however a point that, in my opinion, should be clarified regarding the diffraction image of Fig. 4 (a) for 8 Vpp and circularly polarized incident light. As explained in the Discussion, under this voltage the molecules in the bulk are practically oriented perpendicularly to the glasses, forming a nearly homogeneous medium with uniaxial symmetry (see for example Figs. 8 and 9 in Ref. 25). Thus, the 2D diffraction grating is only consequence of the glass substrates (and the liquid crystal layer close to them). It is argued that the observed diffraction pattern, with hexagonal symmetry, results from the superposition of the effect described in Ref. 40 for each one of the 1D “surface gratings”. Accordingly, three reflections should be observed (1,0) (or (-1,0) but not both), (01) (or (0,-1) but not both), and (1,1). However, the diffraction pattern shows four spots, apart from the (0,0) one. How is this possible?
Other points to be addressed:
In Section 2.2, where the simulation procedure is described, in lines 137 and 138, apart from the elastic constants and dielectric permittivity for E7, the bulk thermotropic coefficients A, B, and C should be defined. A reference should also be given.
Which is the frequency of the AC voltage? Have the authors tested the response with different frequencies?
In line 297 the simulated microscope images are in Figs. 5 and 6 (not in 7, 8).
